# Amylase production from marine sponge *Hymeniacidon perlevis*; potentials sustainability benefits

**Praise Tochukwu Nnaji**[1], **Emmanuel Adukwu**[2‡], **H. Ruth Morse**[2‡], **Rachael U. Chidugu-Ogborigbo**[1,2]*

**1** School of Applied Sciences, College of Health, and Applied Sciences, The University of the West of England, Bristol, United Kingdom, **2** Centre for Biosciences Research, School of Applied Sciences, The University of the West of England, Bristol, United Kingdom

☙ These authors contributed equally to this work.
‡ EA and HRM also contributed equally to this work.
* Rachael.chidugu-ogborigbo@uwe.ac.uk

**Data Availability Statement:** This work is part of my PhD research. I intend to use some of the photographs when writing my final thesis. This thesis will be available online at the University

## Abstract

The marine sponge *Hymeniacidon perlevis* is a globally distributed and invasive species with extensive filter-feeding characteristics. The symbiotic relationship fostered between the sea sponge and the inhabiting microorganism is key in the production of metabolic enzymes which is the focus of this study. Sponge bacterial symbionts were grown on starch agar for 48hrs. Colourimetric analyses of amylase were conducted at 540nm using a spectrophotometric plate reader. Using an X-Bridge column (3.5µM, 4.6x150mm), 80/20 acetonitrile/water in 0.1% ammonium were the conditions used for the liquid chromatography-mass spectrometry (LC-MS) analyses. Seven reducing sugars were used to optimise LC-MS to determine the presence of the crude enzyme formed. Not all the bacterial symbionts isolated from *H perlevis* produced alpha and beta amylases to break down starch. From the statistical mean of crude enzyme concentrations from the hydrolysis of starch by amylase, isolate seven had the highest optical density (OD) at 0.43475 while isolate twelve had the lowest OD at 0.141417. From the LC-MS analysis, out of the seven sugars, Glucose and maltose constituted > 65% of the reducing sugars formed from the hydrolysis of starch by the amylases. Isolates 3,6 and 7 produced 6.906 mg/l, 12.309 mg/l, and 5.909 mg/l of glucose, while isolates 3,4,5,6 and 7 produced 203.391 mg/l, 176.238 mg/l, 139.938 mg/l, 39.030 mg/l, and 18.809 mg/l of maltose, respectively. Isolate two had the highest amount of maltose at a concentration of 267.237 mg/l while isolate four had the highest amount of glucose concentration of 53.084 mg/l. Enzymes from marine sponge bacteria offer greater potential for a green and sustainable production process. Amylase extraction from bacterial symbionts in *H perlevis* is sustainable and should be supported. They can serve as reliable sources of revenue for enzyme industries, and applications in food industries and biotechnological processes.

website https://www.uwe.ac.uk/ The paper contains the minimal dataset.

**Funding:** R.U. Chidugu-Ogborigbo and P.T Nnaji received funding for this work from the University of the West of England, Bristol United Kingdom (https://www.uwe.ac.uk/). Grant Number: RDAS0163. P.T. Nnaji and R.U. Chidugu-Ogborigbo designed the study, P.T. Nnaji collected the data collection,P.T. Nnaji, E. Adukwu, H.R. Morse, and R.U. Chidugu-Ogborigbo analysed the data. P.T. Nnaji prepared the manuscript, all authors agreed to publish. The funders had no role in study design, data collection and analysis, decision to publish, or preparation of the manuscript.

**Competing interests:** The authors have declared that no competing interests exist.

## Introduction

Amylases are a group of enzymes that are extracellularly produced, they are responsible for the breakdown of starch or oligosaccharide molecules into smaller sugars molecules (Fig 1).

They are produced by plants, animals, and microorganisms such as bacteria, fungi and actinomyces etc [1]. Industrially, Amylases are applied in food production processes, pharmaceuticals, detergents, and in textile industries, hence there is an increased demand for amylases in food/enzyme industries. In fact, 25 to 30% of the enzymes sold in the global enzyme market are Amylases [1]. They are grouped into Endo-amylases, exo-amylases, debranching enzymes, and transferases based on their mode of action [2] and are further categorised into alpha amylase, beta amylase, isoamylase and glucoamylase [3]. Alpha amylase is an endo-amylase that catalyses the hydrolysis of glycosidic bonds (alpha-1,4-glycosidic bonds) in starch to produce glucose and maltose [2]. This enzyme contains calcium as a metalloenzyme and depends on a metallic co-factor for its metabolic activities. Different microorganisms, plants, and animals can produce alpha-amylases with the ability to hydrolyse starch [4–6]. The beta amylase (E. C.3.2.1.2) is also produced by various microorganisms, and plants like sweet potatoes, soya beans, ray seeds and higher plants seeds. The activity of beta-amylase involves the hydrolysis of starch to maltose, beta- maltose or beta-limit dextrin [7]. Dextrin units are formed due to the inability of beta-amylase to hydrolyse branched chains of amylopectin and glycogen [3, 4, 7].

The enzyme industries continue to face an increase in the demand for enzymes because of their health, environmental, and economic benefits. Enzymes are also more dependable in bio-catalytic activities than inorganic catalysts [8]. Additionally, enzyme-mediated reaction processes comply with the principles of green and sustainable chemistry more than inorganic catalysts [9–11]. This is because enzyme-based processes offer an environmentally benign catalyst which has selective processes at various levels, and mild or no consequence to the environment at large [11]. They possess high specificity and rate of reaction. Most enzyme-producing industries rely on microorganisms for their production of enzymes. Microorganisms a cultivated at any time and season of the year, they can be engineered and have fast growth and production process compared to plants and animals as sources of enzymes [12]. Microorganisms are more dependable to offer reduced carbon emission, and cleaner and sustainable chemical production processes [13]. However, because of peculiar environmental adaptative traits, marine microorganisms have proven to be one of the most reliable sources of enzymes with higher stability to most extreme conditions [14–16].

Sea sponges are good hosts for marine microorganisms and prominent benthic and coral reef community members. They exist in symbiotic relationships with microorganisms and most benthic aquatic biota. Most sponges possess a simple body plan and rely extensively on getting food from their surrounding environments [17, 18]. Given their extensive symbiotic

**Fig 1. Hydrolysis reaction by amylase in the presence of water molecule.**

relationship with microorganisms, sponges constitute rich sources of key secondary metabolites with unique therapeutic fingerprints [17, 19, 20]. In the Caribbean coral reef, sponges through complex symbiosis with microorganisms participate in key roles like primary production, and nitrification and impact the carbonate framework of the environment [18, 21, 22]. Both the sponge and symbiotic microorganisms produce metabolic enzymes. Sponges are very resilient organisms and are less affected by ocean warming or ocean acidification compared to other benthic organisms, such as corals [19]. They survive and reproduce despite environmental conditions like high salt concentration, low temperatures, elevated temperatures, and high pressure [23]. These properties enlist sponges as excellent sources for the isolation of bacterial enzymes with biotechnological applications in food industries.

In this study, we present data on bacterial symbionts isolated from the sea sponge *Hymenaicidon perlevis* with both alpha and beta amylase activities by colorimetric analysis and LC-MS. *Hymenaicidon perlevis* is a globally distributed sponge species with an all-year-round availability, and to the best of our knowledge, this study is one of the very few to successfully isolate amylases from this sponge species, hence the novelty of this study. Therefore, given the high demand for amylases in the global enzyme market [1], the peculiarity of sponge microbial symbiosis and the need for sustainable enzyme production [11], this study aims to establish a reliable and reproducible methodology for sustained enzyme production from the marine sponge *H perlevis*.

## Materials and methods

### Sample collection and storage

With the aid of a scalpel blade, sponges (Fig 2) were collected randomly during the low tides (~1m) from benthic marine habitats within the coastal rock pools from Tenby Bay (UK). Samples were immediately transported to the laboratory in seawater and immediately preserved until when needed.

### Bacterial culture and staining

With the aid of a Mortar and pestle [24] 5g of sponge tissues (Fig 2) were blended and the juice was extracted with the aid of a sterile pipette. Sponge extracts were serially diluted to $10^{-6}$ following the method described in [25] to isolate the bacteria colonies. Thereafter was the inoculation of 500μl aliquot of extract into starch agar plate by spread and pour plate method at 37˚c. After 48 hours, the bacteria colonies were selected and subcultured/ inoculated on a new starch agar plate to obtain pure colonies. The pure culture was gram stained according to standard gram-staining procedures.

### Starch hydrolysis test

After 48 hours of incubation of bacterial isolates, the hydrolysis of starch was evaluated by flooding pure culture of bacterial growth with an iodine solution. After a few minutes of observation, the zone of clearance around the bacterial colony indicated starch hydrolyses by amylase while the lack of zone of clearance with uniform blue-black colouration indicated the absence of starch hydrolysis.

### Amylase production

Amylase enzymes were produced using the submerged fermentation method from bacterial-positive isolates previously obtained from the starch hydrolysis test. Firstly, using a sterile wire loop each bacterial isolate was inoculated into the nutrient broth and incubated at 37˚c for

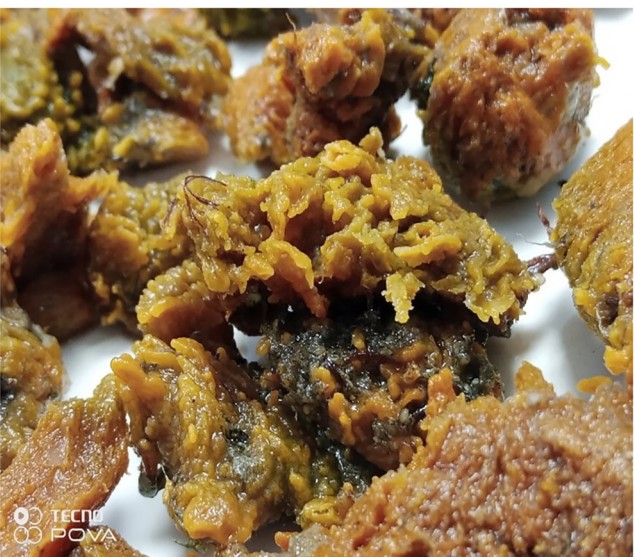

**Fig 2. Sponge samples from the benthic marine habitat on rock pools from Tenby Bay Castle Beach Pembrokeshire, South Wales (Latitude: 51° 40' 22.04" N; Longitude: -4° 42' 16.09" W).**

24hrs. Afterwards, the culture media was vortexed for even distribution of bacterial growth and 1ml of the broth culture was thereafter inoculated into an amylase extraction medium (nutrient broth and 1% soluble starch) to produce amylase. This medium was incubated for 24 to 48 hours at 37˚C, and afterwards, the culture was centrifuged at 8000g at 4˚c for 20 minutes to sediment the bacterial cells. The supernatant containing the crude enzyme was carefully separated from the sediment (bacterial cells) for further analysis.

## Colourimetric enzyme assay

To estimate and quantify the amount of reducing sugars formed, the 3,5-dinitro salicylic acid (DNS) reagent was utilised. The reagent was prepared following the method described in [26]. Approximately 30 g of sodium potassium tartrate and 1g of DNS were suspended in 20 ml of 2M NaOH and the final volume was made up to 100 ml with ultra-pure high-quality (UHQ) water. The mixture was gently heated and mixed on a magnetic stirrer with a heating plate to homogenise the reagents.

Using the methods described in [26–29], the reducing sugars produced by amylase starch hydrolysis were assayed using sugar solutions prepared in UHQ water. To 1 ml of sugar solution, 4 ml of DNS was added in a test tube, which was thereafter transferred into a water bath set at 25˚C. The absorbance was measured at 540nm using Tecan plate reader spectrophotometer. The presence of reducing sugars was confirmed by a change of colour from light amber to deep red/brown colour due to the reduction of DNS to 3-amino-5-nitrosalycylic acid (ANS) (Fig 3).

## Liquid chromatography mass spectrometry

Following crude enzyme extraction by submerged liquid-state fermentation, the presence of reducing sugars was confirmed by liquid chromatography-tandem mass spectrometry methodology using an Agilent 1260 Infinity HPLC system coupled to an Agilent 640 Triple Quadrupole Mass Spectrometer (Figs 4 and 5). The LC-MS mobile phase constituted of 80/20

**Fig 3. Molecular description of DNS reduction reaction with reducing sugar adopted from [30] with modification.**

acetonitrile/water, and 70/30 water/acetonitrile with 0.1% ammonium sulphate in each solution. Seven reducing sugars (Maltose, glucose, sucrose, mannose, fructose, lactose, and galactose) were used as standards to optimise the LC-MS system. The X-Bridge amide 3.5μm, 4.6x150mm column was used to separate the sugars from the sample in mobile phase 80/20 acetonitrile/water with 0.1% ammonium sulphate.

## Characterization of amylase

Amylase was characterised by a reduction reaction evidenced by the change in colour in the colourimetric assay and a confirmation of the presence of different reducing sugars produced by each of the bacterial isolates. The submerged fermentation broth was centrifuged at 8000g for 20mins at 4˚c and the supernatant was carefully removed. The supernatant containing the crude enzyme was analysed using DNS analysis (Fig 3) and LC-MS analysis (Figs 4 and 5).

## Standard curve

To determine the amount of starch reduced during submerged fermentation, a standard curve of two sugars (maltose and glucose) was made by preparing standard solutions of maltose and glucose sugars at a concentration of 50, 100, 150, 200 and 250 mg/l.

### Statistics

All results were analysed in triplicate across three technical repeats using IBM SPSS 28 and Microsoft Excel. Statistical differences between control readings and concentrations of the Isolates were analysed using 1-way ANOVA at a significance level of $P = 0.05$.

## Results

### Starch hydrolysis test

After 48 hours of incubation, the results of the serially diluted crude sponge extract, produced 4 distinct isolates of bacteria colonies. Microscopic examination of the colonies showed varying morphological characteristics of each isolate. The morphological features observed were colours, shape, texture, elevation, and margin (Table 1). The observed colours of the colonies were Milky/creamy white (Isolates 1, 2, and 9); yellow to golden yellow (Isolates 3, 11 and 12); Brown (Isolates 4,5,7,10, and 13). Isolate 1 produced a unique orange-coloured colony. 9

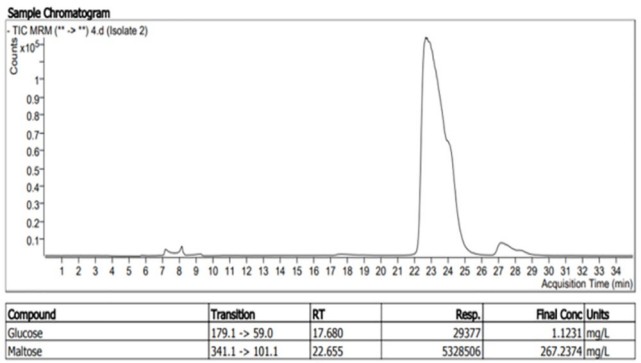

**Fig 4. Chromatograph of maltose as major reducing sugar produced from starch hydrolysis by beta amylase from isolate two.**

(Isolates 1–5, 6,8,10 and 14) of the Isolates had flat elevations. All other isolates had elevations that were recorded as raised. The colonies produced by all the isolates also ranged from irregular to round shapes, except for Isolates 1,2 and 6 whose colonies were filamentous in shape.

Growth of mixed colonies was observed with various morphological characteristics as previously described. Discrete colonies with uniform morphological characteristics were sub cultured into fresh starch agar plates to obtain pure colonies. After flooding the pure colonies with iodine, the hydrolysis of starch was indicated by the zone of clearance around the bacterial colony as indicated below (Fig 6).

## Gram stain

All isolates of pure colonies from the sponge sample were subjected to gram staining. This technique was adopted to verify bacterial organisms. With the Gram staining, bacteria would retain the purple colour from the primary stain (crystal violet) indicating a gram-positive bacterium as indicated in "A and B." On the contrary, gram-negative bacteria will lose the primary stain upon decolourisation and pick up a counter pink colour from safranin as shown in slide "C." (Fig 7).

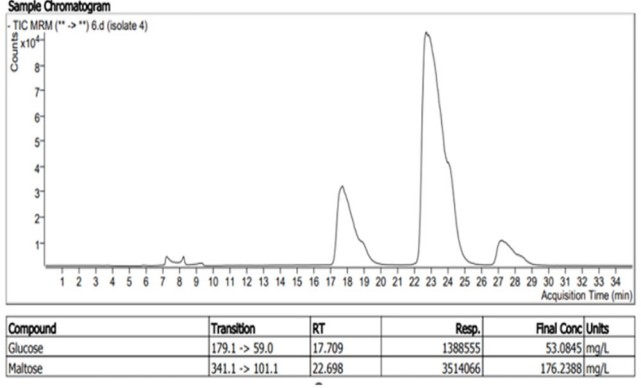

**Fig 5. Chromatographic view of Glucose and maltose; reducing sugar produced from starch hydrolysis by alpha-amylase from isolate four.**

**Table 1. Morphological characteristics of bacterial isolates from *Hymeniacidon perlevis* at different salt concentrations.**

|  | Colour | Shape | Texture | Flat | Margin | Salt Concentration |
|---|---|---|---|---|---|---|
| 1 | Orange | Filamentous | Moist | Flat | Filamentous | 4% |
| 2 | Milky | Filamentous | Moist | Flat | Curled | 4% |
| 3 | Milky | Irregular | Moist | Flat | Curled | 4% |
| 4 | Yellow | Round | Moist | Flat | Smooth/entire | 4% |
| 5 | Brown | Irregular | Moist | Flat | Serrated | 6% |
| 6 | Brown | Irregular | Moist | Raised | Curled | 6% |
| 7 | Opaque | Filamentous | Moist | Flat | Filamentous | 4% |
| 8 | Brown | Irregular | Slimy / Moist | Slightly elevated | Lobate | 4% |
| 9 | White | Flower-like spindle | Dry | Flat | Irregular | 6% |
| 10 | Milk | Irregular | Moist | Raised | Curled | 0% |
| 11 | Brown | Round | Dry | Flat | Entire | 6% |
| 12 | Golden yellow | Round | Moist | Raised | Entire with a ring | 4% |
| 13 | Golden yellow | Round | Moist | Raised | Entire | 4% |
| 14 | Milky brown | Irregular | Moist | Flat | wavy | 4% |

## Colourimetric enzyme assay

The amount of the reducing sugar produced from the hydrolysis of starch by bacterial amylases was analysed by the 3,5-dinitrosalycyclic acid (DNS) method. The colorimetric assay (Fig 8a) and various absorbances at 540nm from twelve bacterial isolates (Fig 10) are represented below.

## Standard curve

To quantify the concentration of glucose and maltose (Figs 9 and 10) formed from the hydrolysis of starch by amylase produced from the sponge bacterial isolate, the standard curve was developed using the X-Bridge amide 3.5μm, 4.6x150mm column for the LC-MS. The graph was plotted using known concentrations of the reducing sugars to check for their corresponding responses.

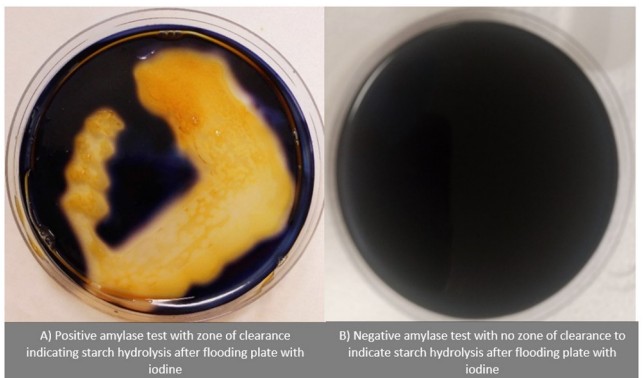

**Fig 6. Photograph of positive (plate A) and negative (Plate B) test for starch hydrolysis by flooding growth in the plate with iodine.**

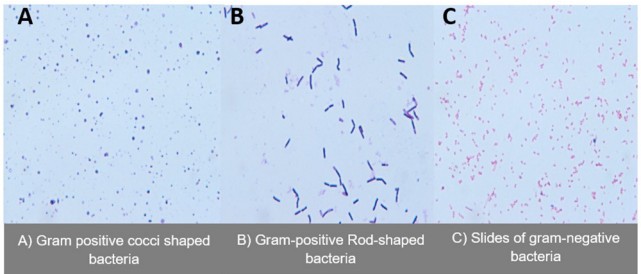

**Fig 7. Slides of gram-positive and negative bacteria from marine sponge *H. perlevis*.**

## Discussion

In this research, sponge-bacterial symbionts were cultured on starch agar medium (a medium used to differentiate amylase producers from non-amylase producers when flooded with iodine). From the growth of bacteria on this medium, and the extraction processes, it is observable that the marine sponge *H perlevis* served as a host for microorganisms of various amylase-producing capacities. Sequel to the isolation of twelve amylase-positive bacteria, (Fig 11) submerged liquid-state fermentation (SLF) technique was used for the extraction of amylase. This

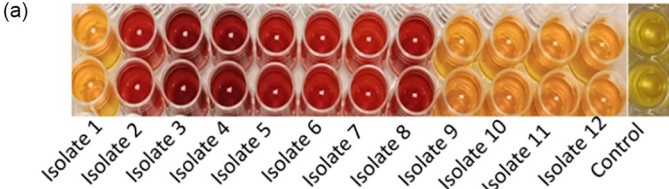

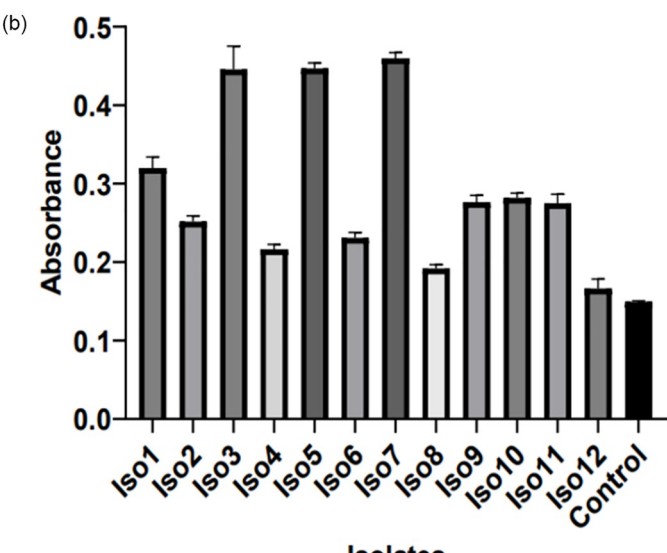

**Fig 8.** a: Reduction reaction indicated by the change of colour from amber yellow to deep brown and a control (DNS and distilled water). b: Graph indicating the quantity of enzyme formed from starch hydrolysis by amylases produced from symbiotic bacteria isolated from *H. perlevis* (n = 3). (Plotted values are mean ± SEM, P<0.05).

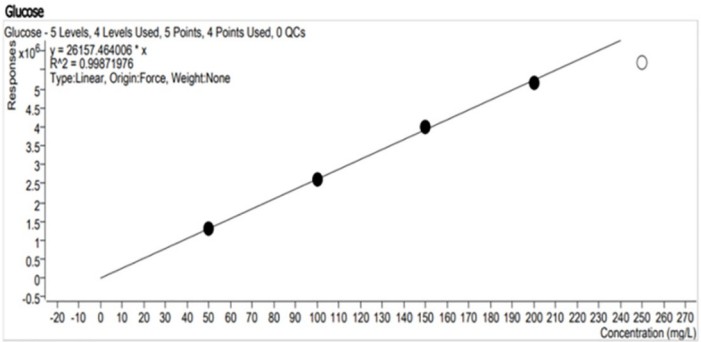

**Fig 9. Standard curve plot from LCMS responses to 50, 100, 150, 200 and 250 mg/l concentrations of glucose in the mobile phase (80/20 acetonitrile/water with 0.1% ammonium).**

technique provided an adequate environment for bacteria to feed, reproduce and release amylase to break down starch molecules into smaller reducing sugars.

The DNS analysis with the aid of a spectrophotometer at 540nm was used to quantify the level of starch hydrolysis in the medium. This method (DNS assay) was adopted since it is the most widely used method for the quantification of reducing sugar [31]. After separating the bacterial cells from the crude enzyme and treating the crude enzyme with DNS, a reduction reaction is expected if the starch has been hydrolysed (Fig 8a). From the results (Fig 8a and 8b) there was evidence that DNS (3,5 dinitrosalicylic acid) had reacted with reducing sugar to produce 3-amino-5-nitrosalicylic acid (ANS). The amount of ANS produced determined how deep or light the colouration of each crude enzyme treatment was when compared to the control. The light amber-orange colour of DNS changes to a deep amber or red-brown colouration due to the production of ANS. The level of reducing sugar produced by the hydrolysis of starch is expected to be directly proportional to the absorbance level i.e. the amount of amylase produced (Fig 8b).

At this point, the reducing sugars produced from the hydrolysis of starch by amylase were unknown. It was therefore necessary to conduct more precise qualitative and quantitative analyses to know exactly which reducing sugars were formed and at what concentrations using liquid chromatography-mass spectrometry. From the LC-MS analysis of the crude enzyme (amylases) produced by the sponge bacterial symbionts, it was observed that out of the seven reducing sugars (glucose, maltose, fructose, galactose, mannose, lactose, and sucrose) which

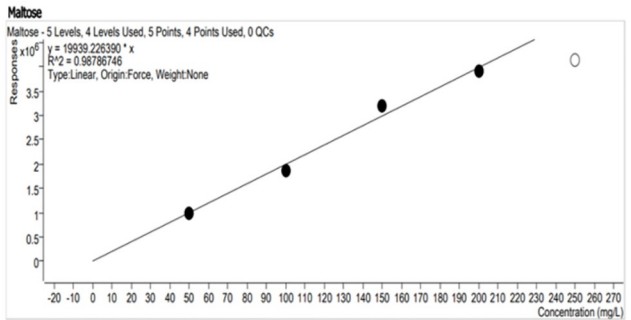

**Fig 10. Standard curve plot from LCMS responses to 50, 100, 150, 200 and 250 mg/l concentration of maltose in mobile phase (80/20 acetonitrile/water with 0.1% ammonium).**

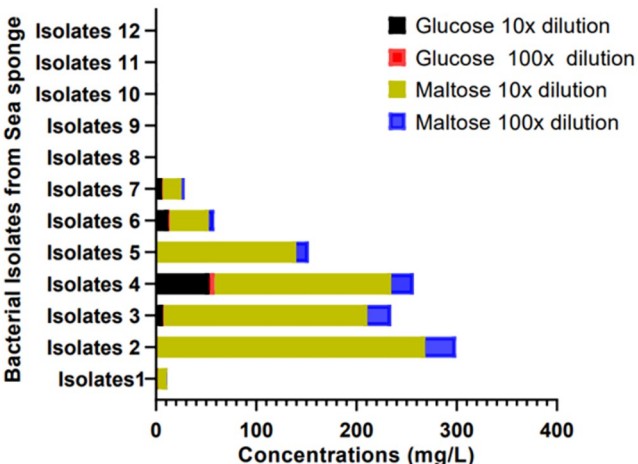

**Fig 11. Glucose and maltose concentration (mg/l) from LC/MS analysis at x10 and x100 dilutions.**

was used to optimise the machine, only glucose and maltose were identified as the major reducing sugars of the starch hydrolysis. The result of the analyses (Fig 11) at a x10⁻¹ dilution, showed that isolates 2 (Fig 4) had the highest concentration of maltose production at 267.237mg/l, followed by isolate three with 203.391mg/l, isolate four with 176.238 mg/l and isolate five with 139.938 mg/l. It was also observed that isolate four had the highest glucose concentration of 53.084 mg/l (Fig 5) followed by isolate six with 12.309mg/l, isolate three with 6.906mg/l and isolate seven with 5.909mg/l.

From these results, (Figs 4 and 11) the formation of maltose following the hydrolysis of starch by amylase suggests that the amylase is a *β*-amylase (EC 3.2.1.2). This result agrees with the findings of [3, 4] and implies that α-1,4-glucan linkages [32] in the starch were hydrolysed by the enzyme to produce maltose. Starch hydrolysis by amylase enzymes from Isolates 3,4,6 and 7 produced a significant amount of glucose and maltose as reducing sugars; this confirms alpha amylase as the amylase produced (E.C.3.2.1.1). This result also agrees with the findings of [2] which signifies that alpha-1,4-glycosidic bonds have been hydrolysed by the enzyme to produce glucose and maltose.

Sponge bacterial symbionts from *H perlevis* is a reliable source for amylase production given the global distribution of the sponge around marine ecosystems [33]. In addition, marine enzymes are beginning to become preferred for biotechnological and other industrial applications [34]. In the view of sustainability, the need for new sources of enzymes to replace inorganic sources cannot be overemphasised. The amylase produced by the marine sponge bacterial symbionts as a biocatalyst for the hydrolysis of starch into glucose, maltose, and other starch moieties, encourages the practice of green and sustainable chemistry. Amylase-producing sponge bacterial symbionts can be isolated within environmentally benign temperature range (10˚c to 37˚c). Both α-amylase (E.C.3.1.1) and *β*-amylase (EC 3.2.1.2) produced by the isolates function with water which is an environmentally benign solvent [35].

Since *H perlevis* is widely distributed along coastal rockpools of marine water bodies, there are chances of exploiting the production of alpha and beta amylases from bacterial symbionts in *H perlevis* for commercial purposes. Sponge tissues serve as host for microorganisms because of the symbiotic relationship that exists between the sponge and microorganisms like bacteria.

## Conclusion

Analysis of the hydrolysed bonds and the resultant sugars formed using LCMS provided matching fingerprint evidence that aligned with those of alpha and beta amylases. Hence, in this study, we have successfully isolated starch hydrolysing bacteria with significant amylase activities from our sea sponge species *H. perlevis*. Fig 6 shows a positive amylase test with a zone of clearance to indicate starch hydrolysis after flooding the plate with iodine. This further confirmed amylase activity in the isolates. To the best of our knowledge, this study is the first to establish amylase activities from isolates of this sponge species.

This research has evidenced the role of the marine sponge *H. perlevis* in sponge bacterial symbiosis. The sponge in this relationship serves as a host for culturable bacterial symbionts with the ability to produce both alpha and beta amylase. Given the global distribution of *H. perlevis* and the ability of the bacterial isolates to produce amylase in a little or nontoxic production medium, therefore, portrays this amylase production process as benign and a process with high sustainability potential.

## Recommendation

Various researchers have proposed enzyme-mediated reactions as the key to green and sustainable chemistry [36]. Enzyme from marine microorganisms is a better way forward for a green and sustainable enzyme-mediated production process.

The use of amylase from the marine sponge *H perlevis* is encouraged as a sustainable practice given the global distribution of the marine sponge, the benefits associated with amylases from extreme environments like the marine environment, the economic potentials and ease of growth. This research supports the sustainable development goal of proper conservation and use of marine bodies and resources for a sustainable development. Achieving green and sustainable processes is not just guaranteed because an enzyme is involved [11]. There is need to monitor enzyme mediated processes, select environmentally benign options, and cost-effective processes to keep the process sustainable.

## Acknowledgments

Appreciation to Lee Graham, James Dawson, Paul Bowdler, and Paul Deane for their technical support in the 2K6 Microbiology lab, UWE Forensic LC/MS lab and School of Applied Science (SoAS) stores.

## Author Contributions

**Conceptualization:** Praise Tochukwu Nnaji, Rachael U. Chidugu-Ogborigbo.

**Data curation:** Praise Tochukwu Nnaji, Rachael U. Chidugu-Ogborigbo.

**Formal analysis:** Praise Tochukwu Nnaji.

**Funding acquisition:** Rachael U. Chidugu-Ogborigbo.

**Investigation:** Praise Tochukwu Nnaji.

**Methodology:** Praise Tochukwu Nnaji.

**Project administration:** Praise Tochukwu Nnaji.

**Software:** Praise Tochukwu Nnaji.

**Supervision:** Emmanuel Adukwu, H. Ruth Morse.

**Validation:** Rachael U. Chidugu-Ogborigbo.

**Writing – original draft:** Praise Tochukwu Nnaji.

**Writing – review & editing:** Praise Tochukwu Nnaji, Rachael U. Chidugu-Ogborigbo.

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
