## [Decision Letter · Decision Letter 0]

12 Jun 2023

PONE-D-23-14066AMYLASE PRODUCTION FROM MARINE SPONGE HYMENIACIDON PERLEVIS; POTENTIALS SUSTAINABILITY BENEFITSPLOS ONE

Dear Dr. Nnaji,

Thank you for submitting your manuscript to PLOS ONE. After careful consideration, we feel that it has merit but does not fully meet PLOS ONE’s publication criteria as it currently stands. Therefore, we invite you to submit a revised version of the manuscript that addresses the points raised during the review process.

We look forward to receiving your revised manuscript.

Kind regards,

Saleh Ahmed Mohamed

Academic Editor

PLOS ONE

“We wish to appreciate the University of the West of England (UWE) Bristol United Kingdom for funding this research. We also appreciate the UWE Forensic LC/MS senior technician and the 2K6 Microbiology Technical team for their technical support.”

“R.U. Chidugu-Ogborigbo and P.T Nnaji recieved funding for this work from the University of the West of England, Bristol United Kingdom (https://www.uwe.ac.uk/).  P.T. Nnaji and R.U. Chidugu-Ogborigbo designed the study, P.T. Nnaji collected the data collection,P.T. Nnaji,  E. Adukwu, H.R. Morse, and R.U. Chidugu-Ogborigbo  analysed the data. P.T. Nnaji prepared the manuscript, all authors agreed to publish.”

“R.U. Chidugu-Ogborigbo and P.T Nnaji recieved funding for this work from the University of the West of England, Bristol United Kingdom (https://www.uwe.ac.uk/).  P.T. Nnaji and R.U. Chidugu-Ogborigbo designed the study, P.T. Nnaji collected the data collection,P.T. Nnaji,  E. Adukwu, H.R. Morse, and R.U. Chidugu-Ogborigbo  analysed the data. P.T. Nnaji prepared the manuscript, all authors agreed to publish.”

Additional Editor Comments:

The authors should be revised the manuscript based on reviewer comments

Reviewers' comments:

Reviewer's Responses to Questions

**Comments to the Author**

1. Is the manuscript technically sound, and do the data support the conclusions?

Reviewer #1: Partly

2. Has the statistical analysis been performed appropriately and rigorously? 

Reviewer #1: I Don't Know

3. Have the authors made all data underlying the findings in their manuscript fully available?

Reviewer #1: No

4. Is the manuscript presented in an intelligible fashion and written in standard English?

Reviewer #1: No

5. Review Comments to the Author

Reviewer #1: the manuscript entitled: AMYLASE PRODUCTION FROM MARINE SPONGE HYMENIACIDON PERLEVIS; POTENTIALS SUSTAINABILITY BENEFITS

deal with using sea sponge and the inhabiting microorganism for the production of metabolic enzymes

general comments:

the manuscript requires English proof reading using a professional service,

the paper deal with production of amylase however quantification and production optimization and characterization

starct is the substrate for alpha amylase enzyme this is not a finding.

characterization of amylase was not included.

citations need to be updated

the main aims of the study should be clearly stated in the last part of the introduction.

other comments are included in the PDF file

here are some specific comments:

introduction: please add literature survey about amylase production from sponge and how your work is different

please add a short paragraph at the end of the introduction summarising the study aims

please refer in literature survey to previous work

updated citations are recommended:

Biotechnology approach using watermelon rind for optimization of α-amylase enzyme production from Trichoderma virens using response surface methodology under solid-state fermentation. Folia Microbiol. (Praha) 67, 253–264. https://doi.org/10.1007/s12223-021-00929-2

Optimization of nano spray drying parameters for production of α-amylase nanopowder for biotherapeutic applications using factorial design. Dry. Technol. 37, 2152–2160. https://doi.org/10.1080/07373937.2019.1565576

Figure2:a picture of the collected sponge is more relative please add and refer to in text

2.2 BACTERIAL CULTURE AND STAINING

The sponge tissue was blended, please explain how?

2.4 AMYLASE PRODUCTION AND EXTRACTION please rewrite to be more clear

2.5 COLORIMETRIC ENZYME ASSAY

as the study is focused around amylase please add more details to the assay and an updated reference

it is confusing to refer to enzyme as product. Please fix through out manuscript

Updated citation for assay methodology is recommended

2.7 LIQUID CHROMATOGRAPHY MASS SPECTROMETRY

please fix anylase for amylase----and through out manuscript

procedure not clear please rewrite

2.8 STATISTICS

please specify the software used is it excel? how many experiments were performed for each experiment, please specify.

3.1 STARCH HYDROLYSIS TEST and GRAM STAIN

what is meant by pure cultures here? please rewrite to be more clear

results are not clear

submerged liquid-state fermentation first mentioned in discussion please refer to in methods

Figure 11 should be included in methods not discussion section

conclusion should summarize the work and the results obtained

the authors did not quantify the Amylase produced? please add

6. PLOS authors have the option to publish the peer review history of their article (what does this mean?). If published, this will include your full peer review and any attached files.

Reviewer #1: No

---

## [Author Response · Author response to Decision Letter 0]

14 Aug 2023

Dear Editor and Reviers,

We thank the reviewers for their positive comments and for their useful critiques which we have now addressed in full as outlined in the "response to reviewers" letter.

Best wishes

Dr Rachael U Chidugu-Ogborigbo

Rachael

Best Regards

Praise Nnaji

---

## [Editor Report · Decision Letter 1]

25 Aug 2023

PONE-D-23-14066R1AMYLASE PRODUCTION FROM MARINE SPONGE HYMENIACIDON PERLEVIS; POTENTIALS SUSTAINABILITY BENEFITSPLOS ONE

Dear Dr. Chidugu-Ogborigbo,

Thank you for submitting your manuscript to PLOS ONE. After careful consideration, we feel that it has merit but does not fully meet PLOS ONE’s publication criteria as it currently stands. Therefore, we invite you to submit a revised version of the manuscript that addresses the points raised during the review process.

We look forward to receiving your revised manuscript.

Kind regards,

Saleh Ahmed Mohamed

Academic Editor

PLOS ONE

Additional Editor Comments:

The authors should be revised the manuscript based on reviewer comments.
---

## [Author Response · Author response to Decision Letter 1]

14 Oct 2023

Dear Rose Ann Joyce Puetes,

Thank you for your latest email. We have now updated the Funding information as:

Financial Disclosure: 

R.U. Chidugu-Ogborigbo and P.T Nnaji received funding for this work from the University of the West of England, Bristol United Kingdom (https://www.uwe.ac.uk/). Grant Number: RDAS0163 

See below and the attached letter detailing response to reviewers comment:

Reviewer 1: We thank referee one for their positive comments and for their useful critiques which we have now addressed in full as outlined below:

Point 1: The manuscript has been proofread.

Point 2: We thank the referee for this comment. The subject of this paper is amylase production from starch hydrolysis by symbiotic bacterial isolates from the marine sponge (H. perlevis) and its sustainability significance. Based on the kind of bonds hydrolysed and the sugars formed from the starch hydrolysis we were able to characterise the type of amylase produced into alpha and beta amylases using LCMS. To the best of our knowledge, no one has researched on the amylase produced by this sponge species, this is novel and a major contribution to this field of study given the cosmopolitan nature of this sponge species. 

Point 3: Lines 185-192 to the end of the discussion contain information on amylase characterisation: Characterization of amylase has been added to the methods. Amylases were characterised as Alpha and beta amylases based on Colorimetric assay and the sugars were identified by LC-MS. This information is in the results and discussion section. See point 17 below for more information.

Point 4: Citations have been updated

Point 5: Line 102-105: The main aim of the study has been updated in the last part of the introduction.

Point 6: Line 29: Bracket has been added to “3.5µm, 4.6x150mm” in line 29

Point 7: The sentence in lines 28-30 has been restructured as requested

Point 8: Lines 49-68: The entire sentence has been rewritten. The main reason for this paragraph, however, is because starch is the major substrate used in this study and serves as an indicator for the presence or absence of amylase in the samples analysed. Hence, we believe it is useful to include a statement that makes the connection between Starch and Amylase to provide some context for readers in the introduction.

Point 9: Lines 56-57: Reference added

Point 10: Lines 59-61: the phrase “Alpha-amylase has the enzyme commission number “E.C.3.2.1.1”. has been removed.

Point 11: Lines 97-105: a short paragraph about the current study has been added

Point 12: Lines 115-117: Figure 2 has been replaced with a picture of the collected sponge and is referred to in the text. 

Point 13: Line 121: Information on how the sponge tissue was blended has been updated.

Point 14: Lines 137-145: The paragraph has been rewritten.

Point 15: The manuscript has been proofread

Point 16: Lines 343-356: Clarification on the novelty of our findings. The subject of this paper is amylase production from starch hydrolysis by symbiotic bacterial isolates from the marine sponge H.perlevis, and its sustainability significance. Analysis of the hydrolysed bonds and the resultant sugars formed using LCMS provided matching fingerprint evidence that aligned with those of alpha and beta amylases. Hence, we are confident that this study successfully isolated starch hydrolysing bacteria from our sea sponge species with significant amylase activities. Fig 4 shows a positive amylase test with a zone of clearance to indicate starch hydrolysis after flooding the plate with iodine. This further confirmed amylase activity in the isolates. To the best of our knowledge, this study is the first to establish amylase activities from isolates of this sponge species. We believe this is novel.

Point 17: Lines 187-192 to the end of the discussion contain information on amylase characterisation: Characterization of amylase has been added to the methods. Alpha and beta amylases were found to be the enzymes produced. This information is in the results and discussion section. The result section contains information on the quantitative characterisation of amylase activities using DNS colourimetric assay. The isolates as can be seen in Figs 5a and b showed significant amylase enzyme activity. This was further confirmed by LCMS and the maximum activity of isolates was achieved in 4% starch and salt concentration.

Point 18: Lines 151-163: Section re-written and more details about Colourimetric assay have been added as requested

Point 19: The term “product” has been replaced with crude enzyme or simple sugar throughout the manuscript.

Point 20: Lines 167-177: Information on Liquid Chromatography Mass Spectrometry procedure/methodology has been updated. 

Point 21: Lines 194- 198: Information on the statistical software used as well as the number of replicates has been updated

Point 22: Lines 201-212: The term “pure culture” has been replaced with “pure colonies”. This session has also been rewritten.

Point 23: Line 137 and Line 167: Submerged liquid fermentation is referred to in the methods session.

Point 23: Figure 11 moved from the discussion to the methods section as requested.

Point 24: Lines 349- 362: The conclusion has been updated with a clearer summary of the work and results obtained.

---

## [Editor Report · Decision Letter 2]

13 Nov 2023

AMYLASE PRODUCTION FROM MARINE SPONGE HYMENIACIDON PERLEVIS; POTENTIALS SUSTAINABILITY BENEFITS

PONE-D-23-14066R2

Dear Dr. Chidugu-Ogborigbo,

We’re pleased to inform you that your manuscript has been judged scientifically suitable for publication and will be formally accepted for publication once it meets all outstanding technical requirements.

Kind regards,

Saleh Ahmed Mohamed

Academic Editor

PLOS ONE

Additional Editor Comments (optional):

The revised manuscript entitled "AMYLASE PRODUCTION FROM MARINE SPONGE HYMENIACIDON PERLEVIS; POTENTIALS SUSTAINABILITY BENEFITS" is acceptable for publication in PLOS ONE
---

## [Editor Report · Acceptance letter]

29 Nov 2023

PONE-D-23-14066R2 

AMYLASE PRODUCTION FROM MARINE SPONGE *HYMENIACIDON PERLEVIS*; POTENTIALS SUSTAINABILITY BENEFITS 

Dear Dr. Chidugu-Ogborigbo:

I'm pleased to inform you that your manuscript has been deemed suitable for publication in PLOS ONE. Congratulations! Your manuscript is now with our production department. 

Kind regards, 

on behalf of

Dr. Saleh Ahmed Mohamed 

Academic Editor

PLOS ONE